# STARJOB: DATASET FOR LLM-DRIVEN JOB SHOP SCHEDULING

## ABSTRACT

The Job Shop Scheduling Problem (JSSP) presents a significant challenge in optimizing production processes. This problem requires efficient allocation of jobs to a limited number of machines while minimizing total processing time (makespan). Although recent advancements in artificial intelligence have produced promising solutions, such as reinforcement learning and graph neural networks, this paper investigates the potential of Large Language Models (LLMs) for addressing JSSP. We introduce the first supervised 120k dataset called Starjob specifically designed to train LLMs for JSSP and we subsequently fintune the LLaMA 8B model on this dataset using Lora. We compare the average makespan gap of our end-to-end LLM-based scheduling method with that of the most widely used priority dispatching rules (PDRs) and neural methods such as L2D. Surprisingly, our findings indicate that LLM-based scheduling not only surpasses traditional PDRs but also achieves on average 11.28% on DMU and 3.29% gap improvement on the Tailard benchmarks compared to the state-of-the-art L2D method.

## 1 INTRODUCTION

The job shop scheduling problem (JSSP) remains a well-studied and computationally challenging problem in the field of production scheduling and optimization. It entails the efficient allocation of a set of $N$ jobs, each with heterogeneous processing times, to a limited number of $M$ machines. The primary objective is to optimize a performance metric, such as minimizing the total completion time (makespan, denoted by $C_{max}$) or reducing the flow time (average completion time) of individual jobs. JSSP finds application in diverse manufacturing and service environments, impacting factors like production throughput, resource utilization, and ultimately, customer service levels. Traditional approaches to JSSP have primarily relied on mathematical programming techniques and heuristic algorithms Chaudhry & Khan (2015). However, these methods often exhibit limitations in scalability and effectiveness, particularly for large-scale problems, or those with complex job-machine precedence relationships. This has motivated the exploration of alternative approaches, particularly with the recent advancements in artificial intelligence (AI). Techniques like reinforcement learning and graph neural networks have shown promise in addressing JSSP, offering data-driven solutions to this problemZhang et al. (2020)Corsini et al. (2024).

Huang et al. (2022) explored the graph reasoning capabilities of large language models (LLMs) through natural language on tasks like connectivity, shortest paths, and more complex challenges such as maximum flow and Hamilton path. LLMs represent a class of AI models trained on massive datasets of text data. While LLMs demonstrate some preliminary graph reasoning abilities, their performance declines with increasing problem complexity, and they often rely on spurious correlations. To enhance performance, Huang et al. (2022) proposed new prompting strategies. Valmeekam et al. (2022) introduce a benchmark to test for evaluating the planning/reasoning capabilities of LLMs. Recently, Chen et al. (2024b) investigate the application of LLMs to the task of graph node classification. Collectively, these studies highlight the growing use of LLMs for tasks involving implicit graphs and structures, though the application of LLMs on scheduling problems still remains largely unexplored. These studies motivated further investigation into testing LLMs capabilities in JSSP. This paper explores the potential of LLMs in tackling the complexities of the JSSP. To the best of our knowledge, we are the first to utilize LLMs for end-to-end scheduling in JSSP problems. We posit that LLMs, with their inherent ability to process and reason over complex information, can be

effectively harnessed to address JSSP. To this end, we introduce the first supervised dataset Starjob [1] designed to fine-tune LLMs specifically for the task of JSSP. Instead of traditional matrix representation format, this dataset includes natural language description of the JSSP problem and solution. On two well-known JSSP benchmarks TaiTaillard (1993) and DMUDemirkol et al. (1998), we show that minimal fine-tuning through RsLoRA Kalajdzievski (2023) on the proposed dataset enables LLM to schedule, by finding high-quality solutions, surpassing PDRs and exceeding or equating neural approaches.

The contributions of this work to the field of JSSP are multifaceted:

- We introduce the first-ever supervised dataset Starjob containing 120,000 instances specifically designed for training LLMs in the context of JSSP

- We investigate the pioneering idea of applying LLMs for JSSP, presenting an end-to-end method for scheduling JSSP using LLMs. This paper underscores the potential of LLMs to address the complexities of JSSP, paving the way for future research and applications in this field.

- We perform a comparative analysis of LLM-based scheduling against four traditional priority dispatching rules (PDRs) Veronique Sels & Vanhoucke (2012): Shortest Processing Time (SPT), Most Work Remaining (MWKR), Most Operations Remaining (MOPNR), and the minimum ratio of Flow Due Date to Most Work Remaining (FDD/MWKR). Additionally, we compare our approach to the state-of-the-art neural method L2D Zhang et al. (2020), highlighting the effectiveness of end-to-end LLM-based scheduling in comparison to existing classical and neural techniques.

## 2 RELATED WORK

JSSP with more than two machines is proven to be NP-hard Garey et al. (1976). As a result, finding exact solutions for JSSP is generally infeasible, leading to the widespread use of heuristic and approximate methods for practical efficiency Cebi et al. (2020). Traditional approaches to solving JSSP have primarily relied on search and inference techniques developed by the constraint programming community Beck et al. (2010). These techniques effectively leverage constraints to define the relationships and limitations between jobs and resources, enabling efficient exploration of feasible solution spaces and the identification of optimal or near-optimal schedules Nowicki & Smutnicki (2005). A widely used heuristic method in real-world scheduling systems is the Priority Dispatching Rule (PDR) Zahmani et al. (2015). PDRs are simple and effective, although designing an efficient PDR is time-consuming and requires extensive domain knowledge.

Recently, approaches utilizing Deep Learning and Neural Networks have gained attention for finding promising solutions to the JSSP Bonetta et al. (2023); Zhang et al. (2020); Corsini et al. (2024). These methods can be broadly categorized into supervised learning and reinforcement learning (RL). Current research in deep reinforcement learning (DRL) is actively focused on developing advanced methods to tackle JSSP. Existing DRL methods typically represent JSSP as a Markov Decision Process (MDP) and learn a policy network based on DRL techniquesZhang et al. (2020).

Large language models (LLMs) are now being applied to a wider range of tasks beyond language processing. In areas like robotics and planning, LLMs have been employed to direct agents through structured environments Huang et al. (2022).

While there are currently no papers that directly address the scheduling of Job Shop Scheduling Problems (JSSP) using LLMs, some notable works explore the potential of LLMs in mathematical reasoning and programming Chen et al. (2023); Wei et al. (2022); Ahn et al. (2024); Yang et al. (2023). Optimization using large language models (LLMs) has gained significant interest in recent years, with several works exploring their capabilities across various domains Yang et al. (2023). The ability of LLMs to understand and generate natural language has opened new possibilities for optimization tasks that were traditionally solved using derivative-based algorithms or heuristic methodsYang et al. (2023). Notably, Chen et al. (2023) have done a comprehensive evaluation of LLMs, incorporating an examination of their performance in mathematical problem-solving. Chen et al. (2023) introduces a novel approach called "Program of Thoughts" (PoT) prompting. Unlike the

---

[1] https://github.com/starjob42/Starjob

Chain of Thoughts (CoT) methodWei et al. (2022), which uses language models to generate both reasoning steps and computations, PoT separates these tasks. PoT uses language models to generate programming language statements for the reasoning steps and then delegates the actual computation to a program interpreter. In Ahn et al. (2024) the authors conduct a comprehensive survey of mathematical problems and corresponding datasets investigated in the context of LLMs. Ahn et al. (2024) examines the spectrum of LLM-oriented techniques for mathematical problem-solving, providing insights into their strengths and weaknesses. Frieder et al. (2024) explores the impact of LLMs on mathematicians' workflows, envisioning changes in research and education through automated assistance and new exploration methods. It provides empirical evidence on LLMs' performance in solving problems and generating proofs, highlighting both successes and failures to give a balanced view of their current capabilities.

More recent works, such as Yang et al. (2023) highlight the potential of LLMs as optimizers, capable of iteratively refining solutions based on a trajectory of previously evaluated solutions. By leveraging the unique strengths of LLMs, such as their natural language understanding and generation capabilities. Paper demonstrates case studies on two fundamental optimization problems: linear regression and the traveling salesman problem. Yang et al. (2023) demonstrates that in small-scale optimization scenarios, LLMs can generate high-quality solutions solely through prompting, sometimes matching or even surpassing the performance of manually crafted heuristic algorithms.

Explorations into using LLMs for graph learning tasks have yielded notable approaches. Huang et al. (2022) noted that LLMs exhibit some initial graph reasoning capabilities, but their performance decreases with problem complexity, Huang et al. (2022) introduced prompting strategies to improve LLMs graph reasoning. Valmeekam et al. (2022) developed a benchmark for assessing the planning and reasoning abilities of LLMs. More recently, Chen et al. (2024b) examined the use of LLMs for graph node classification tasks. Chen et al. (2024a) introduces two pipelines: LLMs-as-Enhancers, where LLMs refine textual data for Graph Neural Networks (GNNs), and LLMs-as-Predictors, where LLMs generate predictions directly from graph structures in natural language. Additionally, Zhao et al. (2024) presents GRAPHTEXT, a method that translates graphs into natural language for LLM-based reasoning. GRAPHTEXT constructs graph-syntax trees for training-free, interactive reasoning, achieving performance on par with or exceeding supervised GNNs through in-context learning, highlighting LLMs' potential in graph machine learning. Together, these studies emphasize the increasing application of LLMs for tasks related to implicit graphs and structures, while their use in scheduling problems remains largely unexamined.

## 3 PRELIMINARY

JSSP is formally defined as a problem involving a set of jobs $J$ and a set of machines $M$. The size of the JSSP problem instance is described as $N_J \times N_M$, where $N_J$ represents the number of jobs and $N_M$ the number of machines. For each job $J_i \in J$, it must be processed through $n_i$ machines (where $n_i$ is the number of operations for job $J_i$) in a specified order $O_{i1} \rightarrow \ldots \rightarrow O_{in_i}$, where each $O_{ij}$ (for $1 \le j \le n_i$) represents an operation of $J_i$ with a processing time $p_{ij} \in \mathbb{N}$. This sequence also includes a precedence constraint. Each machine can process only one job at a time, and switching jobs mid-operation is not allowed. The objective of solving a JSSP is to determine a schedule, that is, a start time $S_{ij}$ for each operation $O_{ij}$, to minimize the makespan $C_{\max} = \max_{i,j}\{C_{ij} = S_{ij} + p_{ij}\}$ while meeting all constraints. The complexity of a JSSP instance is given by $N_J \times N_M$.

## 4 DATASET GENERATION

In order to try to solve the JSSP with LLM, we first need to represent the problem in natural language. To do that, we have to transform the matrix-based representation in standard JSSP format to a human-readable format. See the example in Listing 1.

```
6 6
2 1 0 3 1 6 3 7 5 3 4 6
1 8 2 5 4 10 5 10 0 10 3 4
2 5 3 4 5 8 0 9 1 1 4 7
1 5 0 5 2 5 3 3 4 8 5 9
2 9 1 3 4 5 5 4 0 3 3 1
1 3 3 3 5 9 0 10 4 4 2 1
55.0
```

Listing 1: Job Shop Scheduling Problem instance (ft06)Fisher & Thompson (1963) with $N_J = 6$ and $N_M = 6$. The problem instance begins with the problem size on the first row, followed by the operations for each job. Odd columns list machines, and even columns list durations. The last row indicates the makespan (55.0)

## 4.1 Converting JSSP problem instance to Natural Language: Feature Generation

The approach describes the machines required for each job, providing a job-centric view of the scheduling problem.

- **Initialization:** Begins by introducing the problem, detailing the number of jobs and machines involved.
- **Problem Organization:** Enumerates jobs, specifying the sequence of the corresponding machines, and their respective durations.

```
1
2 Optimize schedule for 3 Jobs (denoted as J) across 3 Machines (denoted as M) to minimize
      makespan. The makespan is the completion time of the last operation in the schedule. Each
      M can process only one J at a time, and once started, J cannot be interrupted.
3
4 J0:
5 M0:105 M1:29 M2:213
6 J1:
7 M0:193 M1:18 M2:213
8 J3:
9 M0:78 M1:74 M2:221
```

Listing 2: Natural Language description of a JSSP instance of size $N_J = 3$ and $N_M = 3$

## 4.2 Zero-shot inference and Label generation

Our choice of LLM is Meta-Llama-3.1-8B-Instruct-bnb-4bit open-source model with 128K context size. Later we will refer this model as Llama3.1 The model is one of the open-source AI models developed by Meta. Llama3.1 is an auto-regressive language model that uses an optimized transformer architecture. The tuned versions use supervised fine-tuning (SFT) and reinforcement learning with human feedback (RLHF) to align with human preferences for helpfulness and safety AI@Meta (2024).

Initially, we considered performing zero-shot inference with the Llama3.1 to solve the JSSP. However, the model consistently produced general descriptions of how to solve the problem instead of actual solutions. Occasionally, it provided partial solutions, however, during each inference time the structure of the provided solution was different, making it hard to parse the solution.

Because the zero-shot inference results were not satisfactory, we decided to finetune the large language model (LLM) using a supervised approach. This required creating a supervised dataset, which included not only the problem formulations in natural language as described in Section 4 but also the solutions.

To generate feasible solutions, we employed Google's OR-Tools. The configuration for the Google's OR-Tools solver was set as follows:

- Maximum time allowed for the solver: `300` seconds.
- Number of search workers: `42`.
- Search branching strategy: `cp_model.AUTOMATIC_SEARCH`.

We have generated approximately 120,000 random JSSP problems of various sizes [2], ranging from 2x2 to 20x20, with the duration of each operation between 5 and 500 units. We created problems

---

[2]https://github.com/starjob42/Starjob

with asymmetric sizes also, such as 3x2 and 10x5, to enhance the model's generalization capability. Overall, the final dataset consists of around 120,000 natural language descriptions of JSSP problems along with their feasible solutions. Since we limited the maximum allowed time for Google's OR-Tools to 300 seconds, the optimality of solutions for problems with $N_J > 10$ and $N_M > 10$ is not guaranteed. The the generated solution is converted to LLM format as described in 4

```
Solution:
J2-M0: 0+78 -> 78, J1-M2: 0+193 -> 193, J0-M0: 78+105 -> 183,
J0-M1: 183+29 -> 212, J2-M2: 193+74 -> 267, J1-M1: 212+18 -> 230,
J1-M0: 230+213 -> 443, J2-M1: 267+221 -> 488, J0-M2: 267+213 -> 480

Maximum end completion time or Makespan: 488
```

Listing 3: Natural Language description of the solution of JSSP problem instance of size $N_J = 3$ and $N_M = 3$

## 5 TRAINING DETAILS

We fine-tuned LLaMA 3.1, an 8-billion-parameter model from Meta, utilizing a 4-bit quantized version to minimize memory usage. We used Rank-Stabilized Low-Rank Adaptation (RSLoRA) Kalajdzievski (2023) with a rank of $r = 64$ and $\alpha = 64$. The model was trained for one epoch, requiring roughly 70 hours and about 30GB of GPU memory.

## 6 EVALUATION

We evaluate fine-tuned LLM on two well known benchmarks TaiTaillard (1993) and DMUDemirkol et al. (1998) and then conduct a comparative analysis of LLM-based scheduling against four traditional priority dispatching rules (PDRs)Veronique Sels & Vanhoucke (2012) and state of the art neural approach L2DZhang et al. (2020). The PDRs include Shortest Processing Time (SPT), Most Work Remaining (MWKR), Most Operations Remaining (MOPNR), and the minimum ratio of Flow Due Date to Most Work Remaining (FDD/MWKR).

At inferce $max\_seq\_length = 20000$ is used and sampling strategy ($do\_sample = True$) with the default hyper-parameters and with $num\_return\_sequences = 10$. Sometimes none of the solutions generated by fintuned Llama were feasible, so we re-run the inference to get feasible solution. During both training and the inference time the model was loaded in $float4$ format. The inference process itself consumes approximately 30GB of memory on the NVIDIA A6000 GPU with $float4$ data type.

### 6.1 OVERVIEW OF JSSP SOLUTION PARSING AND VALIDATION

Following inference, we employ regular expressions to parse the output string generated by the LLM. This process extracts job number, operation number, machine number, start time, duration, end time for each operation, and the makespan value (if present). The validation process involves the following key steps:

1. **Parsing Inputs:** The function parses the `problem_data` and `solution` string to extract jobs, operations, and declared makespan.

2. **Initial Checks:** It verifies the integrity of the inputs, checking for empty solutions and the presence of all required jobs.

3. **Operation Validation:** Confirms that each operation's machine and duration in the LLM output match the expected values from the problem data.

4. **Machine Conflict Check:** Ensures no overlapping operations on the same machine by sorting operations by start time and checking for overlaps.

5. **Job Precedence Check:** Verifies that the end time of one operation is before the start time of the next within the same job, ensuring correct operation order.

6. **Final Validation:** The actual makespan is computed and compared with the declared makespan to confirm the solution's validity.

If all checks pass, the solution is deemed feasible.

## 6.2 Comparative Analysis with Other Neural Approaches

We compared our results with "Learning to Dispatch for Job Shop Scheduling via Deep Reinforcement Learning" (L2D) Zhang et al. (2020), which uses a Graph Neural Network (GNN) and Proximal Policy Optimization (PPO). L2D's method employs a size-agnostic policy network for generalization. We used the network trained on instances with $N_J = 20$ and $N_M = 20$. Table 1 presents the performance comparison of the Llama-Finetuned model on the proposed Starjob dataset against various scheduling methods (L2D, SPT, MWKR, FDD/WKR, MOPNR) on the Tai Taillard (1993) and DMU Demirkol et al. (1998) datasets, focusing on gap percentages relative to optimal solutions makespan. On the Tai benchmark dataset instances with 15 Jobs, 15 Machines, and with 20 Jobs, 20 Machines, finetuned Llama outperforms all other methods. On instances with 20 Jobs and 20 Machines Llama (33.12%) slightly trails L2D (31.60%) but is better than other PDRs. Average Gap: Finetuned Llama (26.57%) is significantly lower than SPT (61.33%), MWKR (57.66%), FDD/WKR (48.86%), and MOPNR (45.88%).

On the DMU benchmark dataset with 20 Jobs and 15 Machines finetuned Llama (25.64%) again demonstrates superior performance against all methods including L2D(38.95 %) Zhang et al. (2020). Finetuned Llama (28.50%) is also notably lower average gap on DMU benchmark dataset instances having 20 Jobs and 20 Machines.

Table 1: Comparison of PDRs against L2D gainist Finetuned Llama on Starjob dataset and the average Gaps on Tai and DMU Benchmark Datasets. The lower the value, the closer the schedule is to the optimal solution, thus representing better performance.

| | | | | TAI Dataset | | | |
|---|---|---|---|---|---|---|---|
| J | M | L2D | SPT | MWKR | FDD/WKR | MOPNR | Llama-Finetuned-Ours |
| 15 | 15 | 25.95 | 54.64 | 56.74 | 47.45 | 44.98 | 19.68 |
| 20 | 15 | 30.03 | 65.24 | 60.65 | 50.57 | 47.97 | 26.91 |
| 20 | 20 | 31.60 | 64.11 | 55.60 | 47.57 | 43.68 | 33.12 |
| **Average** | **Average** | **29.86** | **61.33** | **57.66** | **48.86** | **45.88** | **26.57** |
| | | | | DMU Dataset | | | |
| J | M | L2D | SPT | MWKR | FDD/WKR | MOPNR | Llama-Finetuned-Ours |
| 20 | 15 | 38.95 | 64.12 | 62.14 | 53.58 | 49.17 | 25.64 |
| 20 | 20 | 37.74 | 64.55 | 58.16 | 52.51 | 45.18 | 28.50 |
| **Average** | **Average** | **38.35** | **64.34** | **60.15** | **53.05** | **47.18** | **27.07** |

## 7 Empirical Performance Analysis

In this section, we provide an in-depth comparison of various job scheduling approaches in terms of the gap percentage, which measures the deviation from the optimal solution. The comparison includes several Priority Dispatching Rules (PDRs), a neural approach (L2D), and a fine-tuned Llama model on proposed Starjob dataset. Figure 1 presents the performance on both TaiTaillard (1993) and DMU Demirkol et al. (1998) datasets across various configurations of jobs ($J$) and machines ($M$). The lower the gap percentage, the closer the schedule is to the optimal solution, thus representing better performance.

The five configurations analyzed are:

- $J = 20$, $M = 20$ (Tai dataset)
- $J = 20$, $M = 20$ (DMU dataset)

- $J = 20$, $M = 15$ (Tai dataset)

- $J = 20$, $M = 15$ (DMU dataset)

- $J = 15$, $M = 15$ (Tai dataset)

The *SPT* (Shortest Processing Time) heuristic consistently exhibits the highest gap percentages, exceeding 60% for most problem instances. This is expected since *SPT*, while simple, often fails to account for job-shop constraints in complex problem settings. The *MWKR* (Minimum Work Remaining) and *FDD/WKR* (Flow Due Date/Work Remaining) heuristics, which are more sophisticated than *SPT*, perform moderately better, with gap percentages ranging between 50% and 70%. However, these heuristics are still outclassed by the machine learning-based approaches, likely due to their myopic decision-making, which does not factor in longer-term scheduling impacts.

The L2D Zhang et al. (2020) model, which leverages neural networks for decision-making, offers significant improvements, reducing the gap to the 30%-40% range. This highlights the benefits of learning-based approaches over traditional PDRs, as L2D can implicitly model complex job-shop interactions and adapt to different problem instances. Surprisingly fine-tuned Llama model on Starjob outperforms all pdr methods, consistently achieving gap percentages below 45%. This demonstrates the ability of LLMs to generalize across problem instances, effectively and sometimes even outperforming the specialized neural L2D model.

The results for the DMU dataset with $J = 20$, $M = 20$ mirror those of the Tai dataset (top-middle plot of Figure 1). Here, we observe that traditional PDRs (*SPT*, *MWKR*, *FDD/WKR*) consistently exhibit high gap percentages, with little to no improvement across problem instances . The L2D model once again shows significant improvements over the PDRs, with gap percentages reduced to the 20%-50% range.

Overall, the results highlight that with minimal fine-tuning on the proposed Starjob dataset, not only Llama was able to provide feasible solutions, but also surpass other traditional approaches.

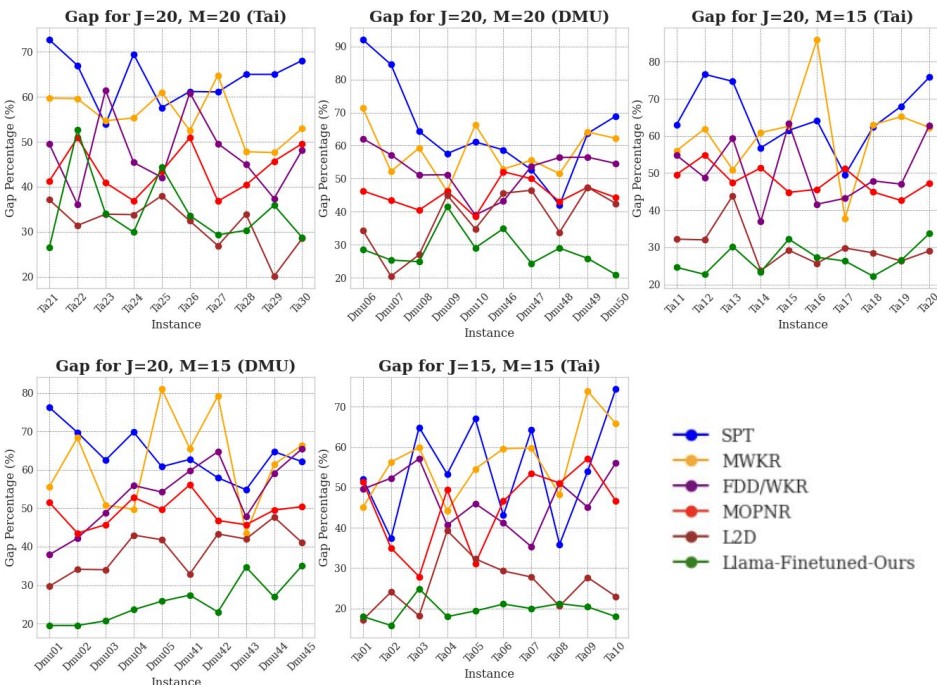

Figure 1: Gap percentage comparison of 4 (PDRs): SPT, MWKR, MOPNR, FDD/MWKR) and neural approach L2D against LLama 8b fintuned on proposed Starjob dataset. The lower the gap percentage, the closer the schedule is to the optimal solution.

## 8  CONCLUSION

This paper demonstrates the potential of Large Language Models (LLMs) in addressing the JSSP. We introduced a novel supervised dataset called Starjob for solving JSSP tailored for LLM training. Our results on well known benchmark problemsTaillard (1993), Demirkol et al. (1998) indicate that with minimal fine-tuning using the RsLoRA methodKalajdzievski (2023), Llama 8B can effectively schedule, matching or surpassing traditional PDRs and neural network approaches.

## 9  LIMITATIONS AND FUTURE WORK

Our exploration of using LLMs for the JSSP marks an important first step in adapting these large models to the domain of scheduling. While the findings are encouraging, they also reveal several challenges and potential directions for future research. The key objective was to experiment with LLMs on scheduling problems like JSSP and demonstrate their initial potential in this domain.

A primary limitation is the significant computational burden associated with fine-tuning and inference of LLMs, which can be quite resource-intensive. Large Language Models (LLMs) are still significantly oversized and resource-intensive when applied to specialized or narrow domain tasks, where a smaller, more efficient model could potentially be more suitable. Furthermore, due to constraints in computational resources, the generalizability of our findings to larger JSSP instances remains uncertain. Consequently, additional research is warranted to assess LLMs on larger problem sizes. It is also crucial to compare the performance of various LLMs and different fine-tuning techniques on the proposed Starjob dataset.

Another challenge lies in the interpretability of schedules generated by LLMs, given their black-box nature. While we employed a basic sampling method to enhance performance, investigating alternative sampling strategies could further improve the quality of LLM-generated schedules.

Looking ahead, future research should consider the integration of LLMs with other artificial intelligence methodologies, such as reinforcement learning and graph neural networks, to leverage their complementary strengths.

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
