# Supplementary Material: Starjob: Dataset for LLM-Driven Job Shop Scheduling

## 1 Trainig Details

### Model Overview

The model being fine-tuned is LLaMA 3.1, an 8 billion parameter model from MetaAI@Meta (2024), using a 4-bit quantized version to reduce memory usage. Finetning was conducted using Stabilized Low-Rank Adaptation (RsLoRA) with rank $r = 64$ to introduce learnable parameters specifically in targeted layers. Kalajdzievski (2023) Compared to LoraHu et al. (2022) RsLoRa improves the stability of training by modifying the rank during adaptationKalajdzievski (2023). The target modules include:

$$\text{target\_modules} = \{\texttt{q\_proj}, \texttt{k\_proj}, \texttt{v\_proj}, \texttt{o\_proj}, \texttt{gate\_proj}, \texttt{up\_proj}, \texttt{down\_proj}\}$$

The LoRA-specific parameters are configured as follows:

- Rank ($r$): 64
- LoRA Alpha ($\alpha$): 64
- LoRA Dropout: 0
- Bias: none

This resulted in number of trainable parameters $= 167,772,160$ or 0.02 % of the entire Llama 8B model's parameters.

### Quantization and Memory Efficiency

The model is loaded in 4-bit precision to reduce memory consumption during training. Gradient checkpointing is enabled using the `unsloth` Unslothai (2024) method, allowing the model to fit longer sequences by saving memory. This reduces the VRAM usage by approximately 30%, enabling larger batch sizes.

### Training Parameters

The fine-tuning process is controlled by the following parameters:

- **Batch size per device**: 4
- **Gradient accumulation steps**: 4
- **Max sequence length**: 10,000 tokens
- **Number of epochs**: 1
- **Warmup steps**: 5
- **Learning rate**: $2 \times 10^{-4}$
- **Optimizer**: AdamW with 8-bit precision
- **Weight decay**: 0.01
- **Learning rate scheduler**: Linear decay
- **FP16 precision**:True
- **Number of Epochs**: 1

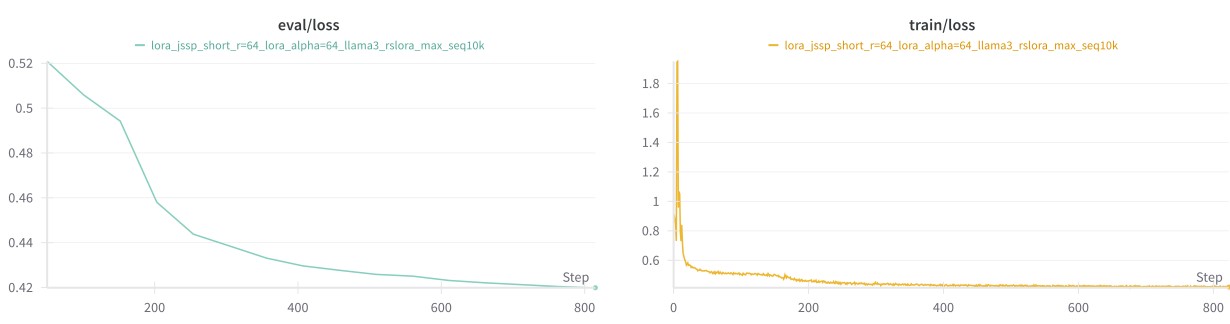

Figure 1: Training and Validation losses of Llama 8B 4bt model on Starjob dataset

## DATA AND DATASET SPLITTING

The dataset used for training is a local version of the proposed Strajob dataset, and it is split into 98% training and 2% evaluation:

$$\text{train : eval} = 98\% : 2\%$$

The prompts are formatted using a predefined Alpaca-style template, which ensures the model is trained on instruction-following tasks.

## EVALUATION AND SAVING STRATEGY

The best model was loaded at the end of training based on the evaluation loss:

$$\text{Metric for Best Model} = \text{Evaluation Loss}$$

Total number of saved models is limited to 50 to prevent excessive memory usage.

## GPU UTILIZATION

The training process takes place on Nvidia A6000 GPU with 48GB of memory. Training took around 70 hours and required 30GB of GPU RAM.

## 2 GENERAL STATISTICS ABOUT DATASET

The dataset is hosted on Github https://github.com/starjob42/Starjob. It includes a data card and detailed information about the dataset, such as various statistics and plots related to makespan, job-machine combinations, and their distribution. The dataset comprises of 120,000 randomly generated JSSP instance problems and their solutions in natural language. It is provided in `.json` format with the following columns:

- `num_jobs` (int64): Number of Unique Values: 12
- `num_machines` (int64): Number of Unique Values: 12
- `instruction` (object): Number of Unique Values: 120,000. Initial description of the problem detailing the number of jobs and machines involved.
- `input` (object): Number of Unique Values: 120,000. Description of the problem in LLM format.
- `output` (object): Number of Unique Values: 120,000. Solution in LLM format.
- `matrix` (object): Number of Unique Values: 120,000. Input problem OR-Tool makespan and solution in Matrix format.

The `output` column serves as the target or label column, providing the solution to the JSSP problem in natural language and the associated makespan.

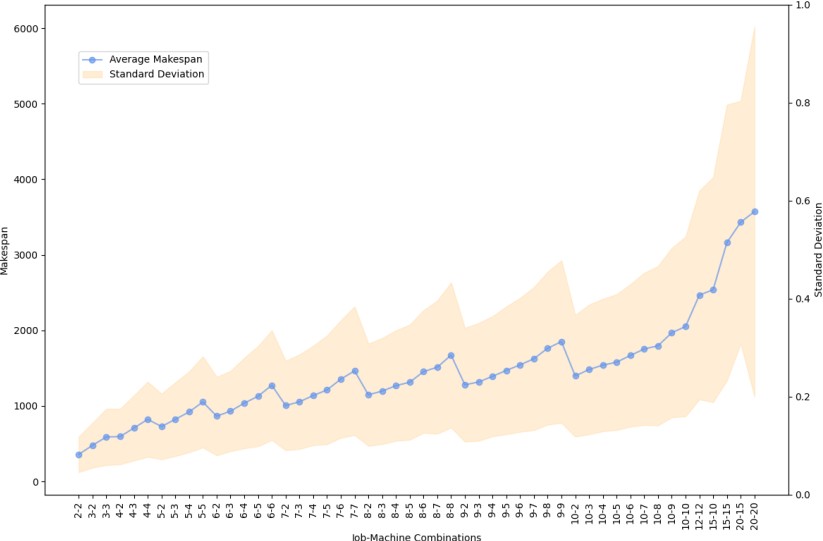

Figure 2: Makespan metrics across different job-machine combinations. The x-axis represents the combinations of jobs and machines (e.g., a 3-2 instance refers to 3 jobs and 2 machines), the right y-axis shows the standard deviation, while the left y-axis shows the makespan values.

## 3    EVALUATION METRICS

Table 1: Comparison of PDRs against L2D gainist Finetuned Llama on Starjob dataset and the average Gaps on Tai Benchmark Dataset. The lower the value, the closer the schedule is to the optimal solution, thus representing better performance.

| J | M | Instance | SPT | MWKR | FDD/WKR | MOPNR | L2D | Optimal | Llama-Finetuned-Ours |
|---|---|---|---|---|---|---|---|---|---|
| 15 | 15 | Ta01 | 1872 (52.1%) | 1786 (45.1%) | 1841 (49.6%) | 1864 (51.4%) | 1443 (17.2%) | 1231.0 | 1453.0 (18.0%) |
| 15 | 15 | Ta02 | 1709 (37.4%) | 1944 (56.3%) | 1895 (52.3%) | 1680 (35.0%) | 1544 (24.1%) | 1244.0 | 1440.0 (15.8%) |
| 15 | 15 | Ta03 | 2009 (64.9%) | 1947 (59.9%) | 1914 (57.1%) | 1558 (27.9%) | 1440 (18.2%) | 1218.0 | 1521.0 (24.9%) |
| 15 | 15 | Ta04 | 1825 (53.3%) | 1694 (44.2%) | 1653 (40.7%) | 1755 (49.4%) | 1637 (39.3%) | 1175.0 | 1387.0 (18.0%) |
| 15 | 15 | Ta05 | 2044 (67.0%) | 1892 (54.6%) | 1787 (46.0%) | 1605 (31.1%) | 1619 (32.3%) | 1224.0 | 1461.0 (19.4%) |
| 15 | 15 | Ta06 | 1771 (43.1%) | 1976 (59.6%) | 1748 (41.2%) | 1815 (46.6%) | 1601 (29.3%) | 1238.0 | 1499.0 (21.1%) |
| 15 | 15 | Ta07 | 2016 (64.3%) | 1961 (59.8%) | 1660 (35.3%) | 1884 (53.5%) | 1568 (27.8%) | 1227.0 | 1473.0 (20.0%) |
| 15 | 15 | Ta08 | 1654 (35.9%) | 1803 (48.2%) | 1839 (51.1%) | 1839 (51.1%) | 1468 (20.6%) | 1217.0 | 1475.0 (21.2%) |
| 15 | 15 | Ta09 | 1962 (54.0%) | 2215 (73.9%) | 1848 (45.1%) | 2002 (57.1%) | 1627 (27.7%) | 1274.0 | 1534.0 (20.4%) |
| 15 | 15 | Ta10 | 2164 (74.4%) | 2057 (65.8%) | 1937 (56.1%) | 1821 (46.7%) | 1527 (23.0%) | 1241.0 | 1465.0 (18.0%) |
| 20 | 15 | Ta11 | 2212 (63.0%) | 2117 (56.0%) | 2101 (54.8%) | 2030 (49.6%) | 1794 (32.2%) | 1357.0 | 1691 (24.6%) |
| 20 | 15 | Ta12 | 2414 (76.6%) | 2213 (61.9%) | 2034 (48.8%) | 2117 (54.9%) | 1805 (32.0%) | 1367.0 | 1677.0 (22.7%) |
| 20 | 15 | Ta13 | 2346 (74.7%) | 2026 (50.9%) | 2141 (59.4%) | 1979 (47.4%) | 1932 (43.9%) | 1343.0 | 1749.0 (30.2%) |
| 20 | 15 | Ta14 | 2190 (56.8%) | 2164 (60.9%) | 1841 (36.9%) | 2036 (51.4%) | 1664 (23.7%) | 1345.0 | 1660.0 (23.4%) |
| 20 | 15 | Ta15 | 2163 (61.5%) | 2180 (62.6%) | 2187 (63.3%) | 1939 (44.8%) | 1730 (29.2%) | 1339.0 | 1770.0 (32.2%) |
| 20 | 15 | Ta16 | 2232 (64.1%) | 2528 (85.9%) | 1926 (41.6%) | 1980 (45.6%) | 1710 (25.7%) | 1360.0 | 1731.0 (27.3%) |
| 20 | 15 | Ta17 | 2185 (49.5%) | 2015 (37.8%) | 2093 (43.2%) | 2211 (51.2%) | 1897 (29.8%) | 1462.0 | 1846.0 (26.3%) |
| 20 | 15 | Ta18 | 2267 (62.4%) | 2275 (63.0%) | 2064 (47.9%) | 1981 (44.9%) | 1794 (28.5%) | 1396.0 | 1706.0 (22.2%) |
| 20 | 15 | Ta19 | 2238 (68.0%) | 2201 (65.2%) | 1958 (47.0%) | 1899 (42.6%) | 1682 (26.3%) | 1332.0 | 1685.0 (26.5%) |
| 20 | 15 | Ta20 | 2370 (75.8%) | 2188 (62.3%) | 2195 (62.8%) | 1986 (47.3%) | 1739 (29.0%) | 1348.0 | 1802.0 (33.7%) |
| 20 | 20 | Ta21 | 2836 (72.7%) | 2622 (59.7%) | 2455 (49.5%) | 2320 (41.3%) | 2252 (37.1%) | 1642.0 | 2077.0 (26.5%) |
| 20 | 20 | Ta22 | 2672 (67.0%) | 2554 (59.6%) | 2177 (36.1%) | 2415 (50.9%) | 2102 (31.4%) | 1600.0 | 2443.0 (52.7%) |
| 20 | 20 | Ta23 | 2397 (53.9%) | 2408 (54.7%) | 2514 (61.5%) | 2194 (40.9%) | 2085 (33.9%) | 1557.0 | 2086.0 (34.0%) |
| 20 | 20 | Ta24 | 2787 (69.5%) | 2553 (55.3%) | 2391 (45.4%) | 2250 (36.9%) | 2200 (33.8%) | 1644.0 | 2135.0 (29.9%) |
| 20 | 20 | Ta25 | 2513 (57.6%) | 2582 (61.0%) | 2267 (42.1%) | 2146 (43.4%) | 2201 (38.0%) | 1595.0 | 2304 (44.4%) |
| 20 | 20 | Ta26 | 2649 (61.2%) | 2506 (52.5%) | 2484 (60.9%) | 2284 (50.9%) | 2176 (32.4%) | 1643.0 | 2195.0 (33.6%) |
| 20 | 20 | Ta27 | 2707 (61.1%) | 2768 (64.8%) | 2514 (49.6%) | 2298 (36.8%) | 2132 (26.9%) | 1680.0 | 2172.0 (29.3%) |
| 20 | 20 | Ta28 | 2654 (65.0%) | 2370 (47.8%) | 2330 (45.0%) | 2259 (40.4%) | 2146 (33.9%) | 1603.0 | 2088.0 (30.3%) |
| 20 | 20 | Ta29 | 2681 (65.0%) | 2399 (47.6%) | 2322 (37.4%) | 2367 (45.7%) | 1952 (20.1%) | 1625.0 | 2209 (35.9%) |
| 20 | 20 | Ta30 | 2662 (68.1%) | 2424 (53.0%) | 2348 (48.2%) | 2370 (49.6%) | 2035 (28.5%) | 1584.0 | 2038.0 (28.7%) |

Table 2: Comparison of PDRs against L2D gainist Finetuned Llama on Starjob dataset and the average Gaps on DMU Benchmark Dataset. The lower the value, the closer the schedule is to the optimal solution, thus representing better performance.

| J | M | Instance | SPT | MWKR | FDD/WKR | MOPNR | L2D | Optimal | Llama-Finetuned-Ours |
|---|---|---|---|---|---|---|---|---|---|
| 20 | 15 | Dmu01 | 4516 (76.2%) | 3988 (55.6%) | 3535 (37.9%) | 3882 (51.5%) | 3323 (29.7%) | 2563.0 | 3064 (19.5%) |
| 20 | 15 | Dmu02 | 4593 (69.7%) | 4555 (68.3%) | 3847 (42.2%) | 3884 (43.5%) | 3630 (34.1%) | 2706.0 | 3233 (19.5%) |
| 20 | 15 | Dmu03 | 4438 (62.5%) | 4117 (50.8%) | 4063 (48.8%) | 3979 (45.7%) | 3660 (34.0%) | 2731.0 | 3296 (20.7%) |
| 20 | 15 | Dmu04 | 4533 (69.8%) | 3995 (49.7%) | 4160 (55.9%) | 4079 (52.8%) | 3816 (43.0%) | 2669.0 | 3299 (23.6%) |
| 20 | 15 | Dmu05 | 4420 (60.8%) | 4977 (81.0%) | 4238 (54.2%) | 4116 (49.7%) | 3897 (41.8%) | 2749.0 | 3458 (25.8%) |
| 20 | 15 | Dmu41 | 5283 (62.7%) | 5377 (65.5%) | 5187 (59.7%) | 5070 (56.1%) | 4316 (32.9%) | 3248.0 | 4137 (27.4%) |
| 20 | 15 | Dmu42 | 5354 (57.9%) | 6076 (79.2%) | 5583 (64.7%) | 4976 (46.8%) | 4858 (43.3%) | 3390.0 | 4169 (23.0%) |
| 20 | 15 | Dmu43 | 5328 (54.8%) | 4938 (43.5%) | 5086 (47.8%) | 5012 (45.7%) | 4887 (42.0%) | 3441.0 | 4634 (34.7%) |
| 20 | 15 | Dmu44 | 5745 (64.7%) | 5630 (61.4%) | 5550 (59.1%) | 5213 (49.5%) | 5151 (47.7%) | 3488.0 | 4429 (27.0%) |
| 20 | 15 | Dmu45 | 5305 (62.1%) | 5446 (66.4%) | 5414 (65.5%) | 4921 (50.4%) | 4615 (41.0%) | 3272.0 | 4423 (35.2%) |
| 20 | 20 | Dmu06 | 6230 (92.0%) | 5556 (71.3%) | 5258 (62.1%) | 4747 (46.3%) | 4358 (34.3%) | 3244.0 | 4173 (28.6%) |
| 20 | 20 | Dmu07 | 5619 (84.5%) | 4636 (52.2%) | 4789 (57.2%) | 4367 (43.4%) | 3671 (20.5%) | 3046.0 | 3821 (25.4%) |
| 20 | 20 | Dmu08 | 5239 (64.3%) | 5078 (59.3%) | 4817 (51.1%) | 4480 (40.5%) | 4048 (27.0%) | 3188.0 | 3982 (24.9%) |
| 20 | 20 | Dmu09 | 4874 (57.6%) | 4519 (46.2%) | 4675 (51.2%) | 4519 (46.2%) | 4482 (45.0%) | 3092.0 | 4376 (41.5%) |
| 20 | 20 | Dmu10 | 4808 (61.1%) | 4963 (66.3%) | 4149 (39.0%) | 4133 (38.5%) | 4021 (34.8%) | 2984.0 | 3853 (29.1%) |
| 20 | 20 | Dmu46 | 6403 (58.7%) | 6168 (52.9%) | 5778 (43.2%) | 6136 (52.1%) | 5876 (45.6%) | 4035.0 | 5447 (35.0%) |
| 20 | 20 | Dmu47 | 6015 (52.7%) | 6130 (55.6%) | 6058 (53.8%) | 5908 (50.0%) | 5771 (46.5%) | 3939.0 | 4899 (24.4%) |
| 20 | 20 | Dmu48 | 5345 (42.0%) | 5701 (51.5%) | 5887 (56.4%) | 5384 (43.1%) | 5034 (33.8%) | 3763.0 | 4854 (29.0%) |
| 20 | 20 | Dmu49 | 6072 (63.7%) | 6089 (64.1%) | 5807 (56.5%) | 5469 (47.4%) | 5470 (47.4%) | 3710.0 | 4674 (26.0%) |
| 20 | 20 | Dmu50 | 6300 (68.9%) | 6050 (62.2%) | 5764 (54.6%) | 5380 (44.3%) | 5314 (42.5%) | 3729.0 | 4515 (21.1%) |

```
### Response:
<|endoftext|>The schedule can be optimized using the NEH algorithm. Here is the optimized schedule:

Machine 1: J0, J2, J5, J7, J9, J11, J13, J15, J17, J19
Machine 2: J1, J3, J6, J8, J10, J12, J14, J16, J18, J20
Machine 3: J4, J21, J23, J25, J27, J29, J31, J33, J35, J37
Machine 4: J6, J22, J24, J26, J28, J30, J32, J34, J36, J38
Machine 5: J7, J19, J21, J23, J25, J27, J29, J31, J33, J35
Machine 6: J8, J20, J22, J24, J26, J28, J30, J32, J34, J36
Machine 7: J9, J21, J23, J25, J27, J29, J31, J33, J35, J37
Machine 8: J10, J22, J24, J26, J28, J30, J32, J34, J36, J38
Machine 9: J11, J23, J25, J27, J29, J31, J33, J35, J37, J39
Machine 10: J12, J24, J26, J28, J30, J32, J34, J36, J38, J40
Machine 11: J13, J25, J27, J29, J31, J33, J35, J37, J39, J41
Machine 12: J14, J26, J28, J30, J32, J34, J36, J38, J40, J42
Machine 13: J15, J27, J29, J31, J33, J35, J37, J39, J41, J43
Machine 14: J16, J28, J30, J32, J34, J36, J38, J40, J42, J44
Machine 15: J17, J29, J31, J33, J35, J37, J39, J41, J43, J45
Machine 16: J18, J30, J32, J34, J36, J38, J40, J42, J44, J46
Machine 17: J19, J31, J33, J35, J37, J39, J41, J43, J45, J47
Machine 18: J20, J32, J34, J36, J38, J40, J42, J44, J46, J48
Machine 19: J21, J33, J35, J37, J39, J41, J43, J45, J47, J49
Machine 20: J22, J34, J36, J38, J40, J42, J44, J46, J48, J50

This schedule minimizes the makespan by assigning the jobs to machines in a way that maximizes the utilization of each machine and mi
nimizes the idle time between jobs. The schedule can be further improved by considering other optimization criteria, such as minimizi
ng the maximum completion time of any machine. However, this schedule provides a good starting point for further optimization. |<|end
oftext|>## Step 1: Understand the problem and the input data.
The problem is to optimize the schedule for 100 jobs (denoted as J) across 20 machines (denoted as M) to minimize the makespan, which
 is the completion time of the last operation in the schedule. Each M can process only one J at a time, and once started, J cannot be
 interrupted.

## Step 2: Read and understand the input data.
The input data consists of 100 jobs, each represented by a list of machines and their corresponding processing times. The jobs are de
noted as J0 to J99, and the machines are denoted as M1 to M20.

## Step 3: Choose an optimization algorithm.
The NEH (Non-Enumerative Heuristic) algorithm is a popular choice for solving the flow shop scheduling problem. It works by iterative
ly constructing a feasible schedule and improving it through a series of exchanges.
```

Figure 3: Zero Shot inference on LLama 8B 4bt