# OpenReview forum: "STARJOB: DATASET FOR LLM-DRIVEN JOB SHOP SCHEDULING"
_ICLR.cc/2025/Conference — ICLR 2025 Conference Withdrawn Submission_

### Official Review · Reviewer_8Xz8 · 2024-11-03

**Soundness:** 2
**Presentation:** 3
**Contribution:** 2
**Rating:** 3
**Confidence:** 3

**Summary:**

StarJob assess how well LLMs can perform the task of Job Shop Scheduling. The authors generate dataset by converting an existing benchmark (Tai and DMU) to an LLM readable format, fine-tune a LLama 8B model on this dataset and demonstrate, that the LLM can perform the JSSP task reasonably well compared to other neural approaches after fine-tuning (at least for a subset of the benchmarking dataset).

**Strengths:**

Proper evaluation of LLMs is an open problem, specifically when it comes to reasoning. There are multiple angles: a) and most of the traditional approaches to assess NLP models are not good evaluation metrics/criteria to assess proper reasoning b) models often have been trained on published benchmarking datasets, and there is a lack of problem diversity.

This paper introduces a new problem domain into LLM evaluation that requires proper reasoning/optimization, and has objective and quantifiable target outcomes that are clearly separated from style of the generated output.

**Weaknesses:**

Limited novelty. The papers main contributions in the generation of a JSSP dataset for LLM evaluation including fine-tuning a 8B LLama model. This overall feels like a narrow contribution. For an evaluation paper highlighting capabilities of LLMs, I would have expected to see a more comprehensive evaluation of JSSP and related problems. For a method paper, I would have expected to see more novelty rather than just fine-tuning a single LLM (or better SLM as the model used is fairly small)

I would like to at least some of the following additions:
1. Assessment over a larger range of models to contrast their capabilities, e.g. other SLMs, proper LLMs such as GPT models (in this case only via prompt engineering, not fine-tuning), etc.
2. I have reservations how this approach would scale to larger JSSP problem sizes. Evaluation is only performed over subset of the available benchmark datasets. I would like to see at least some analysis and discussion on scaling behaviour with JSSP problem complexity. The authors list this under limitations.
3. A wider range of reasoning tasks in the job scheduling domain.

**Questions:**

I would be curious to understand how the method compares to non-neural approaches, both, in terms of achieved accuracy as well as computational cost of LLM based JSSP, other neural JSSP, and non neural approaches.

Do you have a feeling on how much of the fine-tuning is for learning the representation of the problem, vs really improving the problem solving capabilities. Have you experimented with other approaches, e.g. prompt engineering?

---

### Official Review · Reviewer_NaGB · 2024-11-03

**Soundness:** 2
**Presentation:** 2
**Contribution:** 2
**Rating:** 3
**Confidence:** 4

**Summary:**

This paper investigates the potential of Large Language Models (LLMs) for addressing JSSP. To generate labels to train LLM, authors employed Google’s OR-Tools in 300s to collect feasible solutions. Lora adapted LLM with the collected solutions and JSSP problem descriptions. The LLM achieved performance better than PDRs and L2D in TAI, DMU Dataset but suffers long inference time.

**Strengths:**

This paper investigates the potential of Large Language Models (LLMs) for addressing JSSP. The trial is good to exhibit the difficulty to finetune an LLM for a reasoning task, JSSP that seems quite hard by LLMs but can be done efficiently by heuristics and L2D.

**Weaknesses:**

Motivation of applying LLM to a simple task is not reasonable. If a supervised dataset is accessible, it’s more reasonable to train a neural network for the task, rather than forcibly converting the problem in natural language and finally parse solutions back. It takes too much extra time in inference and training. LLM is good at aligning task descriptions so that different LLM foundation models were developed for downstream tasks. The trained LLM only applicable to JSSP does not make sense to me. That is, training an extremely heavy model for a single task does not deserve the effort put in. Section 6.1 parsing procedure didn’t harmonize LLM, which introduced much heuristics.

Related work missed too much recent work of DRL techniques to tackle JSSP. L2D is definitely not a SOTA model currently. Comparison to more recent work is suggested. LLMs for optimization work are missing. The solved problems are small and not practical. Inference time is not reported in the paper.

**Questions:**

1 "At inferce max seq length = 20000 is used and sampling strategy (do sample = True) with the default hyper-parameters and with num return sequences = 10." The hyperparameter definitions were not elaborated. What is their meaning?

2 a. How are proposed techniques used to tackle larger JSSPs? b. How do the trained LLM apply to JSSPs described by texts differing from the format in the paper?

---

### Official Review · Reviewer_Qyiv · 2024-11-04

**Soundness:** 2
**Presentation:** 2
**Contribution:** 1
**Rating:** 3
**Confidence:** 5

**Summary:**

This paper proposes a dataset for training LLM to solve traditional job shop problems (JSSP) containing 120K data samples. LLaMA 3.1 8B is chosen as the supervised-finetuned LLM with RSLoRA and  4-bit quantisation techniques for saving memory. The whole idea of the paper is straightforward. The JSSP instance is represented by natural language. Since the LLM is prone to generate infeasible solutions, the paper uses sampling to get feasible solutions.

**Strengths:**

1. It is interesting to see an LLM finetuned with JSSP data represented in natural language actually has better performance than a neural-based solver (L2D).
2. The paper is organised in a clear structure that is easy to follow. The intuition and the method is easy to understand even for readers outside the field of combinatorial optimization.

**Weaknesses:**

1. The evaluation of the proposed method needs to be stronger. The baselines are relatively simple: mainly are dispatching rules and a neural-based method surpassed by many existing methods
2. LLM is prone to suffering from hallucinations. Therefore, not feasible solutions can be guaranteed at all time. This is the main drawback of using LLM for solving CoP problems.
3. The size of the evaluation is too small, e.g., in L2D, the largest size is 100 x 20.
4. No running time cost comparison is given for the proposed method and baselines.

**Questions:**

1. It would be interesting to see the comparison between not fine-tuned Llama and the Llama fine-tuned with the dataset proposed in the paper. How much is the improvement by using the proposed dataset?
2. I would like to see the training curves.
3. Sampling is not optimal for handling hallucinations (i.e., infeasible solutions). Do you have better ways?
4. How are 120K training data samples distributed across problem sizes?
5. I would like to see the generalisation of the method, e.g., training on small sizes and testing directly on large sizes.

---

### Official Review · Reviewer_D8mA · 2024-11-08

**Soundness:** 3
**Presentation:** 3
**Contribution:** 2
**Rating:** 3
**Confidence:** 2

**Summary:**

The paper introduces Starjob, a dataset designed to fine-tune large language models (LLMs) for solving the Job Shop Scheduling Problem (JSSP). JSSP is a complex optimization task requiring efficient allocation of jobs to machines while minimizing the makespan (total processing time). The authors demonstrate the potential of LLMs in scheduling, specifically by fine-tuning the LLaMA 8B model on their dataset using LoRA (Low-Rank Adaptation). Their LLM-based scheduling approach is benchmarked against priority dispatching rules (PDRs) and a neural method (L2D), showing superior performance in reducing the makespan. The paper presents a novel application of LLMs for end-to-end scheduling in JSSP, with the potential to exceed traditional and neural approaches.

**Strengths:**

**Novel Application of LLMs**: This is the first work applying LLMs to JSSP, pushing the boundaries of LLM applications beyond traditional language processing tasks. The concept of fine-tuning an LLM on a scheduling problem is innovative.

**Dataset Contribution**: The introduction of the Starjob dataset, consisting of 120,000 natural language descriptions of JSSP problems and their solutions, is a valuable resource for future research. It bridges the gap between optimization tasks and natural language models.

**Performance Evaluation**: The paper provides thorough comparative analyses, demonstrating that the LLM-based approach significantly outperforms traditional PDRs and improves upon neural methods in certain benchmarks. The reported improvements in average makespan gap are notable: 11.28% on the DMU benchmark and 3.29% on the Taillard benchmark.

**Interpretability of Data**: The transformation of matrix-based JSSP data into human-readable natural language format for LLM training is a clever approach that enhances the model’s interpretability and generalization ability.

**Weaknesses:**

**Computational Complexity**: The fine-tuning and inference stages are computationally intensive, requiring significant GPU resources (30GB) and long training times (70 hours for one epoch). This limits the accessibility and scalability of the approach, particularly for larger JSSP instances.

**Generalization Concerns**: The model is only for JSSP. I do not know what the general audience could learn anything from this paper.  It seems someone could also train new models for many other indivisual  problems. What is the specicial part of JSSP to a general ICLR audience?

**Questions:**

- Does the problem have the standard optimal answer, for example using an external solver? If so why should we use LLMs for it. The only difference is the consumed time, is it in this case?

- Does the model could be applied to a problem that has a much larger scale?

- How does this paper contribute to a geneneral ICLR audience, or any specical groups of ICLR community?

---

### Note · Authors · 2024-11-28

I have read and agree with the venue's withdrawal policy on behalf of myself and my co-authors.